# Numerical Modeling of Cavitation Rates and Noise Acoustics of Marine Propellers

Kwanda Mercury Dlamini, Vuyo Terrence Hashe [ID] and Thokozani Justin Kunene *[ID]

Department of Mechanical and Industrial Engineering Technology, University of Johannesburg, Doornfontein Campus, Corner of Siemert and Beit Streets, Johannesburg 2028, South Africa
* Correspondence: tkunene@uj.ac.za; Tel.: +27-11-556-6978

**Abstract:** The study numerically investigated the noise dissipation, cavitation, output power, and energy produced by marine propellers. A Ffowcs Williams–Hawkings (FW–H) model was used to determine the effects of three different marine propellers with three to five blades and a fixed advancing ratio. The large-eddy Simulations model best predicted the turbulent structures' spatial and temporal variation, which would better illustrate the flow physics. It was found that a high angle of incidence between the blade's leading edge and the water flow direction typically causes the hub vortex to cavitate. The roll-up of the cavitating tip vortex was closely related to propeller noise. The five-blade propeller was quieter under the same dynamic conditions, such as the advancing ratio, compared to three- or four-blade propellers.

**Keywords:** advancing ratio; noise acoustics; sound pressure level; vortex shedding

## 1. Introduction

The propeller is one of the most prominent producers of noise in the marine environment, and modern propeller designs must meet both hydrodynamic and hydroacoustic requirements [1–3]. In order to protect the environment and save operating expenses during challenging maritime economic cycles, energy efficiency must also be improved [4]. One of the most challenging problems is cavitation. When vapor cavities start to form, there are several unforeseen repercussions on the system, including noise, decreased performance, vibrations, and wall degradation [5]. Cavitation should be minimized through the design and selection of propellers. Cavitation will be the primary source of radiated noise and might considerably increase underwater noise. The objective of a skewed shape in fixed-pitch blades for different maritime propellers is to avoid cavitation while retaining propeller efficiency [6]. Radiated noise and near-field pressure changes are caused by fluid flow processes such as cavitation, turbulence, vortex shedding, displacement, and lift [1]. Sound design, such as reducing propeller load and providing as uniform a water flow through the propellers as feasible, can prevent cavitation under normal operating conditions [7].

In terms of the comfort of mariners and the preservation of the marine environment, underwater ship-radiated noise (USRN) has garnered considerable interest from the maritime sectors. There are noise regulations for fishery research vessels because high underwater noise levels may also affect fish behavior. Underwater ship noise mainly consists of tonal and wideband noise produced by marine propellers, and low frequency and periodic machinery noise from primary and auxiliary engines [1,2]. Tonal blade rate and wideband noise make up the underwater radiated propeller cavitation noise. While sheet cavitation raises noise levels in the low-frequency zone, tip-vortex cavitation significantly contributes to the broadband noise in the high-frequency and medium–low-frequency regions [3].

The application of the Ffowcs Williams–Hawkings (FW–H) approach in conjunction with viscous computational fluid dynamics (CFD) to hydroacoustic problems, namely,

propeller-radiated noise, is rapidly progressing [8]. Far-field noise can be computed using an acoustic analogical method represented by the FW–H wave propagation equation in conjunction with viscous CFD. In the subject of aeroacoustics, the FW–H approach is commonly used [9]. Using a wave equation to simulate the propagation of pressure/density disturbances (noise) in the far field is the main benefit of this hybrid approach to the problem. Advanced CFD solvers such as the large-eddy simulation (LES), detached-eddy simulation (DES), or Reynolds-averaged Navier–Stokes (RANS) can be used to identify the noise source. Thus, the acoustic calculations are released from the computational mesh's spatial constraints [10]. RANS was utilized by Lindau et al. [11] to forecast the cavitation-related breakdown of propeller thrust and torque. The critical cavitation number and the anticipated performance breakdown agreed with the experimental results. For the prediction of cavitation on a propeller, Bensow and Bark [12] combined the incompressible LES model with an implicit modeling approach for the sub-grid term. The flow was regarded as a two-phase, single-fluid combination. In addition to the LES equations, a model transport equation for the local volume fraction of vapor is solved. A finite rate mass transfer model is employed for the vaporization and condensation processes [12].

Smaller, more isotropic sub-grid structures are modeled in LES while larger, energy-containing structures are resolved on the computational grid. This division of scales within the flow is accomplished via (implicit) low-pass filtering of the Navier–Stokes equations. Because it enables medium- to small-scale transient flow configurations spontaneously and reliably, LES is more appealing than RANS [13]. Ji et al. [14] were successful in simulating the unsteady cavitating flows around the INSEAN E779A propeller and achieved some attractive results. Internal jets and leading-edge desinence, two fundamental cavitation mechanisms reported by the authors, may help assess cavitation erosion. Since the vast majority of naturally occurring flows and almost all engineering applications involve turbulence, CFD research primarily focuses on flows in which turbulence dominates [15].

This study made use of the ILES model along with the FW–H formulation in the low-frequency range. An implicit model based on treating the flow as a single fluid, the two-phase mixture, was utilized for the sub-grid term to anticipate cavitation on a propeller. The model LES is then formalized to ILES (Implicit LES). Kimmerl et al. [16,17], with an emphasis on representing likely sound sources, used implicit LES to analyze the tip- and hub-vortex cavitating flows. The same cavitating conditions were used to perform grid sensitivity tests to bring our numerical results to a converged state. For each conventional propeller, the noise intensity and directivity in the near (above the propeller) and far fields are analyzed to establish the cause-and-effect relationship with the properties of the cavitation and sources.

## 2. Numerical Methods

### 2.1. Governing Equation and the Large-Eddy Simulation Turbulence Model

The conservative form of the filtered equations for the conservation of mass and momentum in a Newtonian incompressible flow is given below [15]:

$$\partial_i \overleftarrow{u}_i = 0 \tag{1}$$

$$\partial_t(\rho \overleftarrow{u}_i) + \partial_j(\rho \overleftarrow{u}_i \overleftarrow{u}_j) = -\partial_i \overleftarrow{p} + 2\partial_j(\mu \overline{S}_{ij}) - \partial_j(\tau_{ij}) \tag{2}$$

$$\overline{S}_{ij} = \frac{1}{2}\left(\partial_i \overleftarrow{u}_j + \partial_j \overleftarrow{u}_i\right) \tag{3}$$

$$\tau_{ij} = \rho\left(\overline{u_j u_j} - \overleftarrow{u}_i \overleftarrow{u}_j\right) \tag{4}$$

where $\overline{S}_{ij}$ is the filtered, or resolved scale strain rate tensor, $\rho$ is density, $\overleftarrow{u}_i$ is the filtered velocity, $\overleftarrow{p}$ is the filtered pressure, $\mu$ is the molecular viscosity, and $\tau_{ij}$ is the unknown sub-grid scale (SGS) stress tensor. They are the motions in the resolved fields of LES.

SGS Modeling

Many different types of SGS models have been created, and most use Boussinesq's hypothesis and the eddy-viscosity assumption to represent the SGS stress tensor. They are presented as follows [18]:

$$\tau_{ij} = 2\mu_t \bar{S}_{ij} + \frac{1}{3}\delta_{ij}\tau_l \tag{5}$$

where $\mu_t$ is called SGS eddy viscosity, and we substitute this into Equation (5), which then becomes:

$$\partial_t(\rho \bar{u}_i) + \partial_j(\rho \bar{u}_i \bar{u}_j) = -\partial_i \bar{p} + 2\partial_j[(\mu + \mu_t)\bar{S}_{ij}] \tag{6}$$

Due to the introduction of the modified pressure $\bar{P} = \bar{p} + \frac{1}{3}\tau_{ll}$, when the equation is calculated, the pressure obtained is not merely the static pressure. The only issue left is how to calculate the SGS eddy viscosity, and the simplest model is the one that Smagorinsky initially put forth:

$$\begin{cases} \mu_t = \rho(C_S\bar{\Delta})^2 S \\ S = (2\bar{S}_{ij}\bar{S}_{ij})^{\frac{1}{2}} \\ \Delta = (\Delta x \Delta y \Delta z)^{\frac{1}{3}} \end{cases} \tag{7}$$

where $C_S$ is the so-called Smagorinsky constant which depends on the type of the flow. A Smagorinsky constant of $C_S = 0.1$ was used for this study to account for the near-the-wall effects.

### 2.2. Cavitation Model

The mass transfer equation for the conservation of the liquid volume fraction that governs the cavitation process can be written as follows [19]:

$$\frac{\partial(\rho_l \alpha_l)}{\partial t} + \frac{\partial(\rho_l \alpha_i u_j)}{\partial x_j} = \dot{m}^+ + \dot{m}^- \tag{8}$$

where the phase change's condensation and evaporation rates are denoted by $\dot{m}^+$ and $\dot{m}^-$. The Rayleigh–Plesset equation-derived cavitation model by Kubota et al. [20] is applied in this study. Equation (9) presents the bubble's expansion and deflation as follows:

$$\frac{dR_B}{dt} = \sqrt{\frac{2(p_v - p)}{3\rho_I}} \tag{9}$$

where $R_B$ is the spherical bubble's radius. The model's source and sink terms are then specified as follows:

$$\dot{m}^- = -C_{\text{dest}}\frac{3\alpha_{nuc}(1-\alpha_v)\rho_v}{R_B}\left(\frac{2}{3}\frac{p_v - p}{\rho_I}\right)^{1/2}, p < p_v \tag{10}$$

$$\dot{m}^+ = C_{prod}\frac{3\alpha_v\rho_v}{R_B}\left(\frac{2}{3}\frac{p - p_v}{\rho_l}\right)^{1/2}, p > p_v \tag{11}$$

where $p_v$ is the saturated vapor pressure, $\alpha_{nuc}$ is the nuclei volume fraction, $C_{\text{dest}}$ is the constant generation rate of vapor in the region where the local pressure is less than the vapor pressure, and $C_{prod}$ is the constant rate for re-conversion of vapor back to liquid in a region where the local pressure exceeds the vapor pressure. According to Zwart et al. [21], the model constants are: $\alpha_{\text{nuc}} = 5 \times 10^{-4}$, $R_B = 1 \times 10^{-6}$, $C_{\text{dest}} = 50$, and $C_{\text{prod}} = 0.01$. The validation of the cavitation model with the assumed constants was performed by Huang et al. [22].

*2.3. Hydroacoustic Model: Theory for Acoustic Fan Source [19]*

In 1969, Ffowcs Williams and Hawkings devised the FW–H equation to account for moving solid and permeable surfaces based on Lighthill's [23] acoustic analog; the equation can be expressed as follows:

$$\frac{\partial^2 p'}{\partial t^2} - c^2 \frac{\partial^2 p'}{\partial x_i^2} = \frac{\partial^2 T_{ij}}{\partial x_i \partial x_j} - \frac{\partial}{\partial x_i}\left[P_{ij}\delta(f)\frac{\partial f}{\partial x_j}\right] + \frac{\partial}{\partial t}\left[\rho_0 u_i \delta(f)\frac{\partial f}{\partial x_j}\right] \tag{12}$$

where the sound pressure is defined as $p' = p - p_0$, $c$ is the sound velocity, $f$ is the function of the moving boundary, and $f(x,t) = 0$ and $\delta(f)$. is the Dirac function The stress tensor $P_{ij}$ and the Lighthill stress tensor $T_{ij}$ can be expressed as:

$$T_{ij} = \rho u_i u_j + P_{ij} - c^2 \rho \delta_{ij} \tag{13}$$

$$P_{ij} = p\delta_{ij} - \mu\left(\frac{\partial u_i}{\partial x_j} + \frac{\partial u_j}{\partial x_i} - \frac{2}{3}\frac{\partial u_k}{\partial x_k}\delta_{ij}\right) \tag{14}$$

When the Mach number is low, this work ignores the quadrupole sound source represented by the first component on the right of Equation (15), which is produced by turbulent fluctuations and the interaction of the shear layers. The second term is the dipole sound source, also referred to as the loading noise, which is significant in determining propeller noise. The monopole sound source, also known as thickness noise, is the third term to the right of the equation and is brought on by the blade rotation. As per the theory for an acoustic fan source, the axial and tangential components of the radiated sound pressure can be expressed as:

$$p_{inc}^{mB}(\vec{x}) = \frac{-imB^2\Omega e^{-imB\Omega R/c_0}}{4\pi c_0 R} \times \sum_{s=-\alpha}^{\alpha} F_{sV} e^{-i(mB+sV)(\varphi+\pi/2)} J_{mB+sV}\left(-mBMa \sin\theta\right)\left[\cos\gamma - \frac{mB+sV}{mBMa}\sin\gamma\right] \tag{15}$$

$$p_{inc}^{mB}(\vec{x}) = \frac{-imB^2\Omega e^{-imB\Omega R/c_0}}{4\pi c_0 R} \sin\theta \sum_{s=-\infty}^{\infty} F_{sV} e^{-i(mB+sV)(\varphi+\pi/2)} J'_{mB+sV}\left(-mBMa \sin\theta\right) \tag{16}$$

where $R$ is the distance between the acoustic measurement point and the center, $c_0$ is the sound velocity, $m$ is the harmonic number, $B$ is the blade number, $\Omega$ is the rotating speed, $F_s$ is the Fourier series of the force imposed on the blade segment, and $Ma$ is the rotational Mach number. Consequently, the loading force on the blade may be determined from the pressure fluctuation in the pressure field derived by the hydrodynamic model.

*2.4. Acoustics Analogy*

The Fast Fourier Transform (FFT) algorithm converts overall acoustic pressure measurements from the time domain to the frequency domain for each receiver. The equation below can be used to determine sound pressure level (SPL) [24].

$$SPL = 20\log\left(\frac{p_{rms}}{p_{\text{ref}}}\right) \tag{17}$$

Here, $p_{rms}$ is the root mean square of sound pressure and, $p_{\text{ref}}$ is the reference pressure for $10^{-6}$Pa for water.

## 3. CFD Methodology

The numerical models, computational domain, and grid structure utilized to solve flow around a cavitating propeller with its acoustic analogy are described in this section.

### 3.1. Computational Domain and Mesh Generation

Figure 1 shows the geometries of the model propellers. They were arbitrarily chosen only to suit marine sizes, closely related screw angles, and the number of blades. The numerical calculations kept the propeller rotational speed constant at 300 rpm. The inflow velocities were suitable for corresponding with the advance coefficient of $J = 0.7$ ($J = V_0/nd$: where $n$ = *number of blades*, $V_0$ = *inlet velocity, and d = propeller diameter*). Figure 2 shows the computational domain, with their propeller diameters (d, for each propeller) and rotating domain (D = 2 m), respectively. The length of the main domain is 50D. The distance between the inlet plane and the propeller is 5D. Previous numerical simulations have adopted these parameters in the same field of study [24–26]. Table 1 shows the modeling parameters used for the three types of marine propellers.

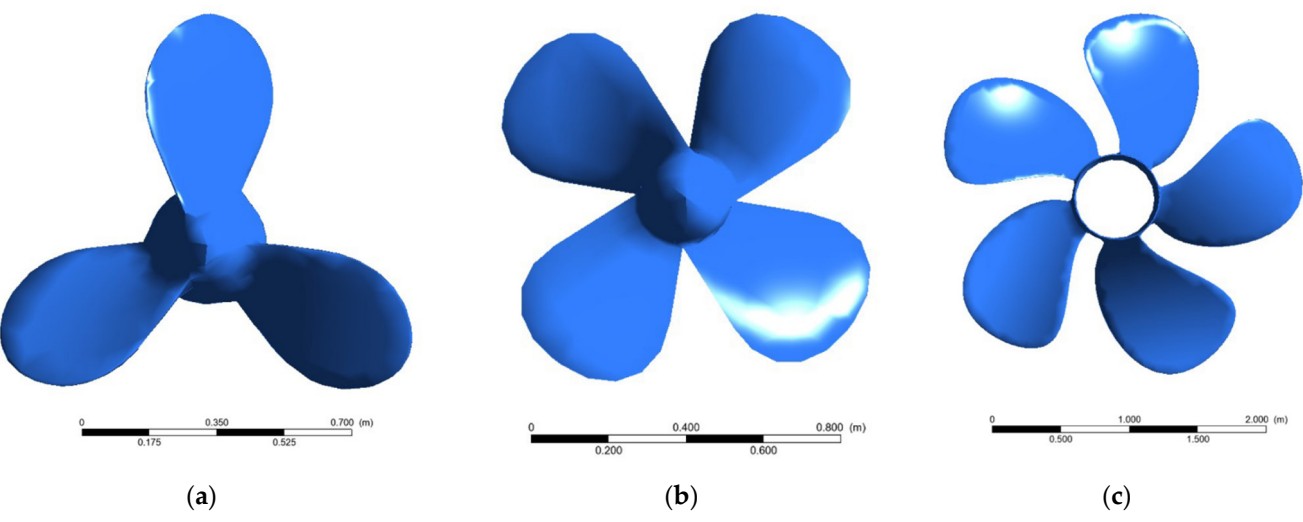

| (a) | (b) | (c) |

**Figure 1.** Topology of fixed-screw-angle marine propellers: (**a**) 3 blades; (**b**) 4 blades; and (**c**) 5 blades.

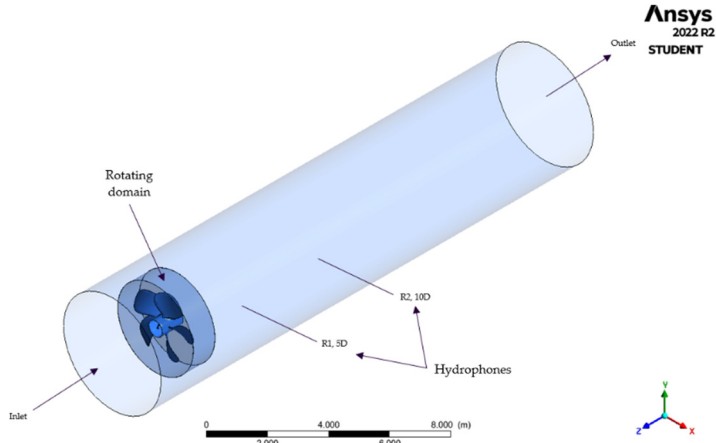

**Figure 2.** Computational domain, boundary conditions, and hydrophone positions.

**Table 1.** Geometric parameters of the propeller and the working conditions. All propellers were modeled at a constant speed of 300 rpm.

| Number of Blades | 3 | 4 | 5 |
|---|---|---|---|
| Propeller diameter, d [m] | 1.1 | 1.1 | 1.5 |
| Skew angle | 0° | 0° | 5° |
| Cavitation coefficient, σ [-] | 7.06 | 6.71 | 1.65 |

Figure 2 shows the observer positions where the hydrophones are located to capture sound pressure were placed at distances 5D and 10D at an axial direction to the propeller. It is the distance from where noise is emitted (propeller) to where the observer receive the noise [27]. This approach was chosen to determine how each propeller type would significantly contribute to a noise signature. The observer positions are not far out in the field but relatively close to the rotating frame. It is assumed that noise would dissipate further downstream.

However, the interest of this work is to link the noise traveled to a distance, to know the quietest propeller. The range-beam of the hydrophone looks at the more comprehensive frequency range of 10 to 80,000 Hz in connection with sonar detection [28]. The flow field of the propeller was solved using the multiple reference frame (MRF) techniques. The MRF technique runs a steady flow simulation at cruise in the rotating frame. The MRF technique improves computational efficiency, essential for the optimization program, by avoiding costly transient flow simulation [22].

A mesh created using a tetrahedron and seven inflation layers with a first thickness of $10^{-5}$ and a wall plus value of $y^+ = 5$ are shown in Figure 3a. There is a 510,000-element maximum limit on the student version of ANSYS® Fluent 2022 R2. It archived 422,660 elements and meshed almost to the limit. Based on the restrictions of the commercial code, these components were accepted as being appropriate for the model. However, a mesh sensitivity study was conducted using ANSYS Workbench Parameterization module (See Appendix A). The element size was varied (progressively reduced) as the cell-mesh size enlarged and are show in Figure 3b. The outlet pressure and the propeller thrust were selected to as quantities that characterize the mean flow These quantities were compared to the cell-meshes until they could not change any more, especially, within the limits of the software.

### 3.2. Solver

The hydrodynamic flow field was solved using an LES solver. The Rayleigh–Plesset cavitation model based on an equation (Equations (10)–(13)) was used to model sheet cavitation on the propeller blades. For the momentum equations, water characteristics were combined with pressure, and the mass fluxes flow. The semi-implicit method for pressure-linked equation (SIMPLE)-type solution algorithm was imposed on the calculations. Temporal and spatial discretization was performed using a second-order scheme to increase the accuracy of the solution. A temporary converged timestep of $1.0 \times 10^{-4}$ was reached for a full revolution of the propeller. The timestep is approximated to be equal to one degree (i.e., 1 timestep $\cong 1^0$ degree of rotation) [29,30].

### 3.3. Model Validation

The current study lacks experimental data, which is only supported by comparable CFD studies that have already been published. Figure 4 shows the validation of the current study with the literature findings. The numerical model was compared to experimental findings [31], a propeller with a diameter of 2.8 m, five blades (skew angle of 5°), and a rotating speed of 163 rpm. The hydroacoustic performance of the propellers under cavitation conditions is compared and indicates similar trends and relative magnitude of propeller noises. Hydroacoustic pressures were discovered in the near field to be in good agreement with one another, as with studies in the literature. The deviance of literature experimental findings from the numerical studies is conjectured to due to the limitation of element sizes (422,660). In addition, the vorticity shedding prevents a full periodic averaging over a revolution, and the simulated sound pressure level (SPL) displays higher values. In contrast, the simulation could be more practical because the experimental results are evaluated by averaging over several revolutions [32].

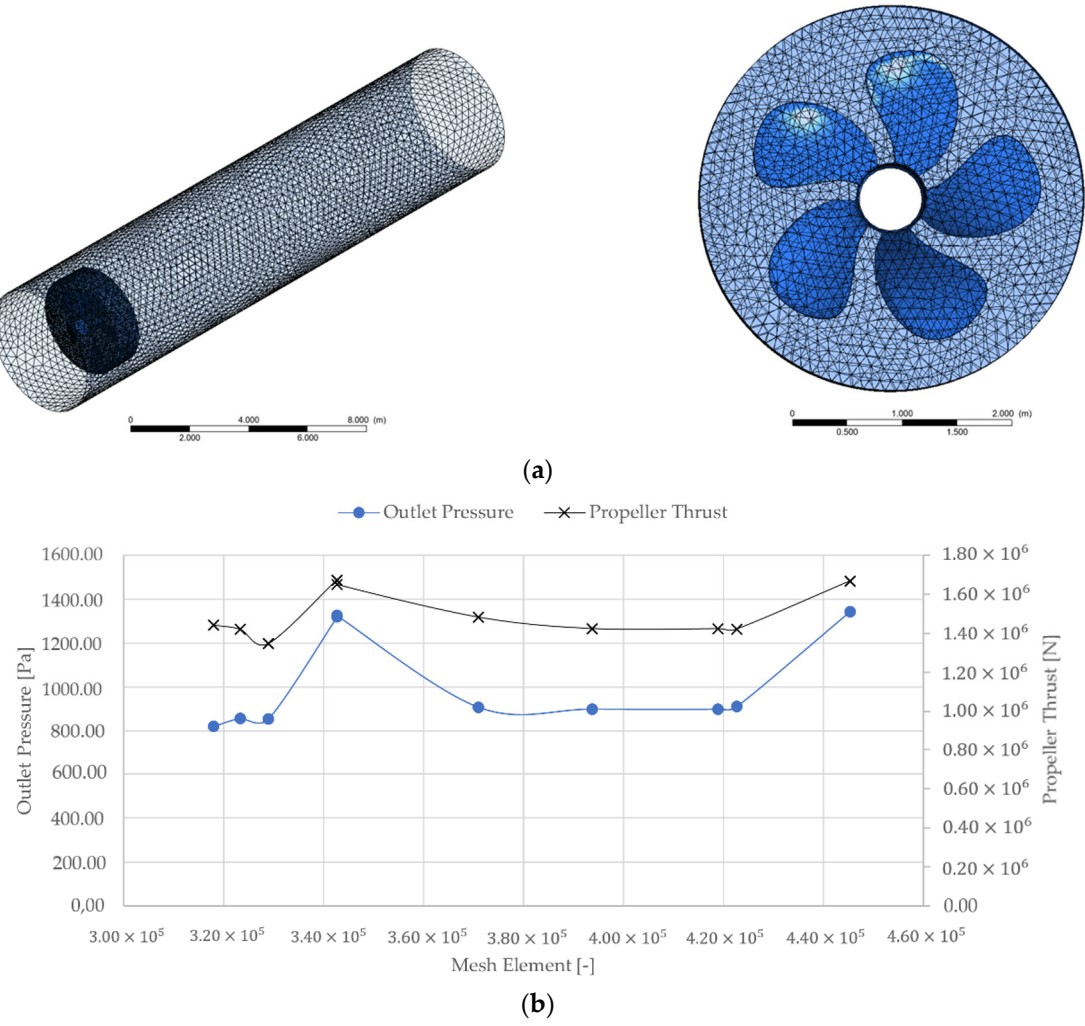

**(a)**

**(b)**

**Figure 3.** (**a**) Tetrahedron grid distributions on the computational domain; (**b**) mesh sensitivity study.

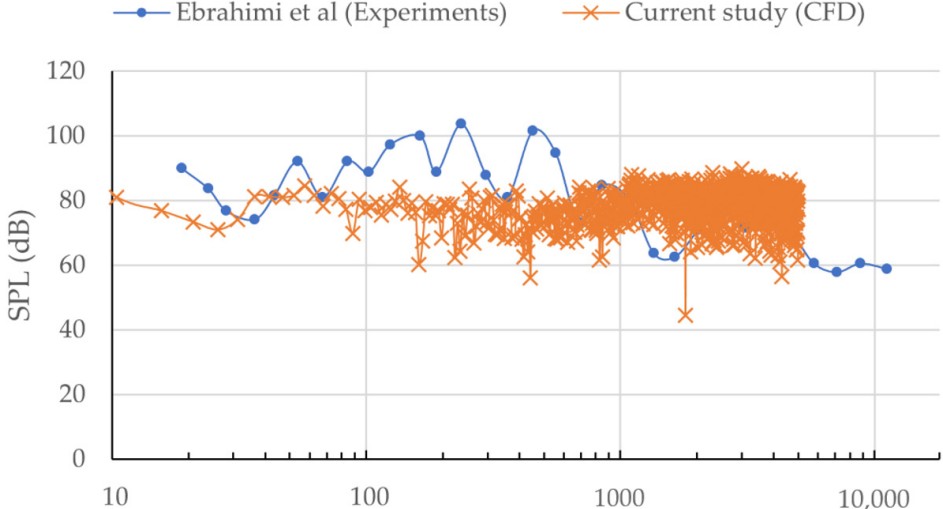

**Figure 4.** Propeller noise validation of the current model compared to literature finding, J = 0.7. Reproduced with permission from Ebrahim et al. [31].

It should be noted that these results only consider the effects of the far-field FW–H equation for the analysis of the cavitation tunnel and do not consider the effects of the computational domain's reflection. As a result, the results of the numerical SPL's equation,

verified in Figure 4, can be used as free-field results. However, a cavitation tunnel is a completely reverberant environment, and wall reflections impact the propeller's overall noise performance [33].

## 4. Results

### *4.1. Vorticity*

The planes on 0d and 5d are situated near the sheet cavity, and the tip-vortex region is shown separately in Figure 5a–c, representing the blades being studied. The plane on the 0d distance depicts the tip vortex on the blade; it is seen that there is a larger vorticity magnitude (18,666.566 1/s) on the tips from the three blades, and it is undoubtedly less on a five-blade propeller (2678.698 1/s). The four-blade propeller exhibits very low magnitudes (197.980 1/s) that, when coupled with the vortex structure's dynamic behaviors driven by cavitation, the vapor volume fraction testifies to the point that cavitation has a significant impact on the vorticity distribution and its transport.

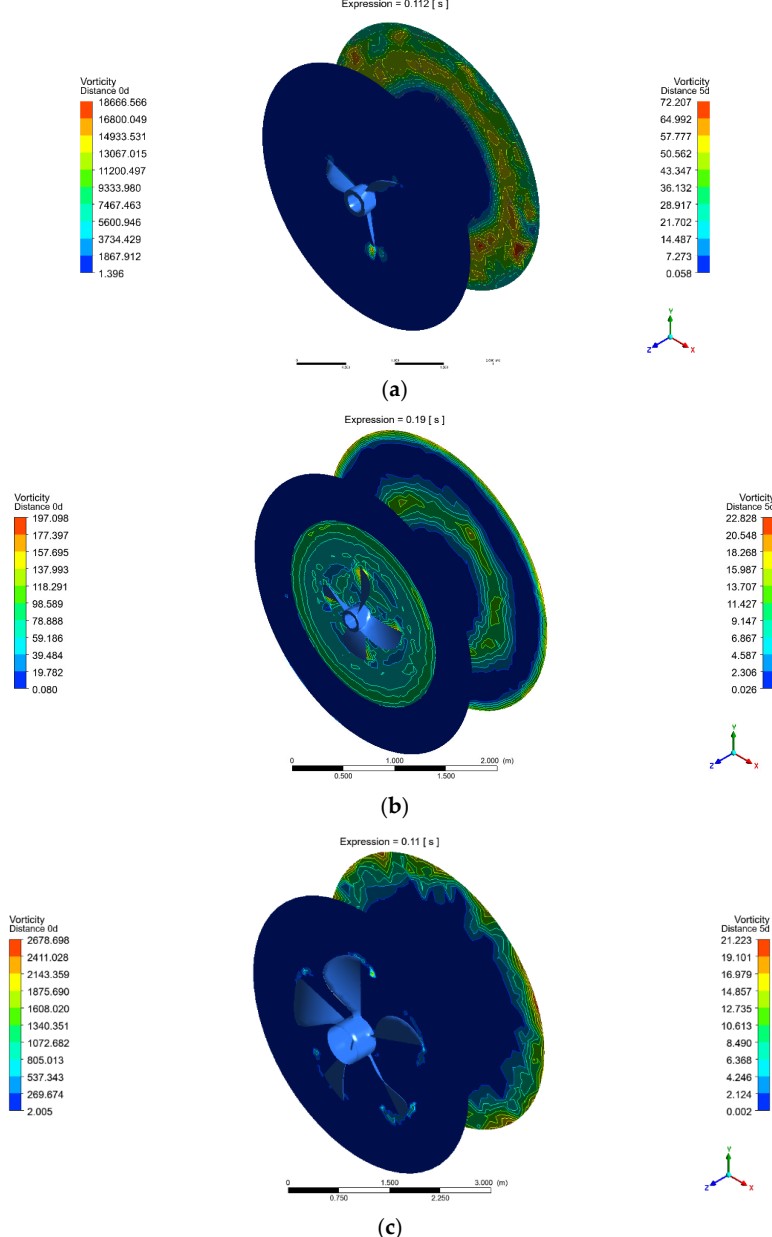

**Figure 5.** Vorticity magnitude in the vorticity region (i.e., tip-vortex region) as a source of sound on the blade and 5d downstream the propeller: (**a**) three blades, (**b**) four blades, and (**c**) five blades.

Two systems of vortex structures make up the propeller's wake and are primarily produced at the blade's root and tip sections. The pressure difference between the blades' face and back sides causes the tip vortices to form. Due to the non-constant circulation and axial hub vortex, a sheet of trailing vortices can be considered additional vortex structures [24].

Higher relative fluid blade velocities, angular attack ranges, and more substantial vorticity sheds take precedence. The separation becomes dominant at higher harmonics with the help of displaced fluid—the spread of the signal and noise. The four-blade propeller's higher noise level indicates the most increased fluid blade fluctuations and angle of attack variations.

### 4.2. Cavitation

The wake's presence significantly impacts the propellers' cavitation and noise performance. The tip-vortex cavitation and leading-edge suction-side sheet cavitation are present and can be used to predict radiated noise levels. This can be performed for all propeller geometries. The results, therefore, are shown in Figure 6.

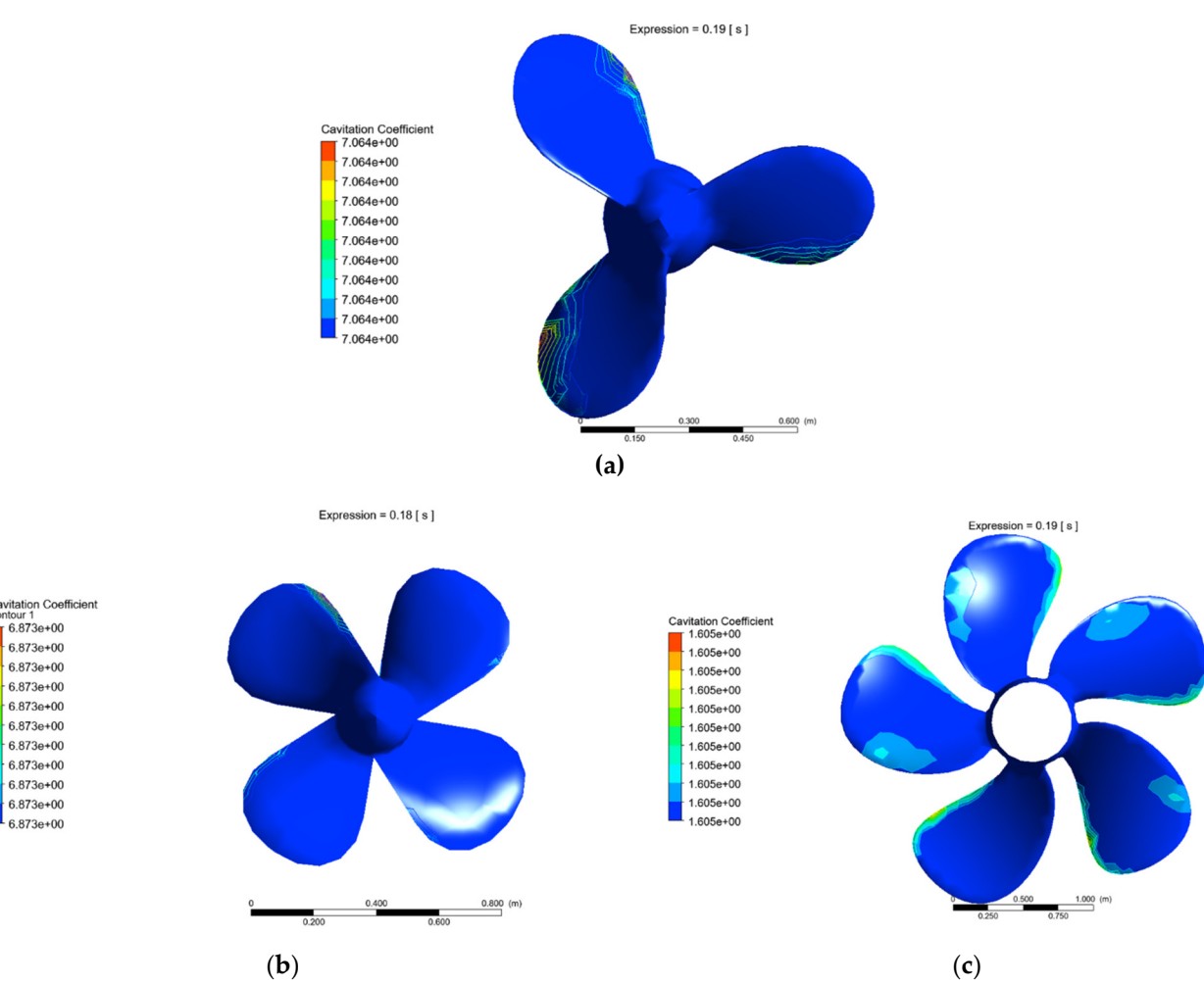

**Figure 6.** The contour visualized the cavitation patterns at 0.19 s for: (**a**) three blades, (**b**) four blades, and (**c**) five blades.

The leading edge separation is more critical in cavity initiation here and at higher angles of attack than in the design [11]. It is seen that the cavitation coefficient decreases on the five-blade propeller, but the tip-vortex cavitation becomes unstable. If the cavitation coefficient reduces any further, surface cavitation develops fully. Larger skew angles lead to

less cavitation [34]. With the increase in the skew angle, the lengths of the sheet cavitation bubbles reduce in the chord-wise direction, demonstrating quieter blade types.

### 4.3. Velocity Profiles

The streamwise velocity fluctuation $V_{th}$, normalized by the inlet velocity $V_0$ and $u$ is the velocity in the *x*-direction, is shown in Figure 7. The low-velocity core with the maximum peak velocities on either side of the jet centerline is present at the 5D stream of the propeller and is typically symmetrical. The three-blade propeller exhibits very high-velocity peaks. The four- and five-blade propellers decrease as the number of blades increases. The velocity profiles of the inlet and outlet for all blade cases are normal distributions of velocity flow for a turbulent flow. Where $z$ is the axial distance and D is the diameter of the rotating domain. A note is made that the main flow domain was not rotating but only the rotating domain where propellers have their various impressions.

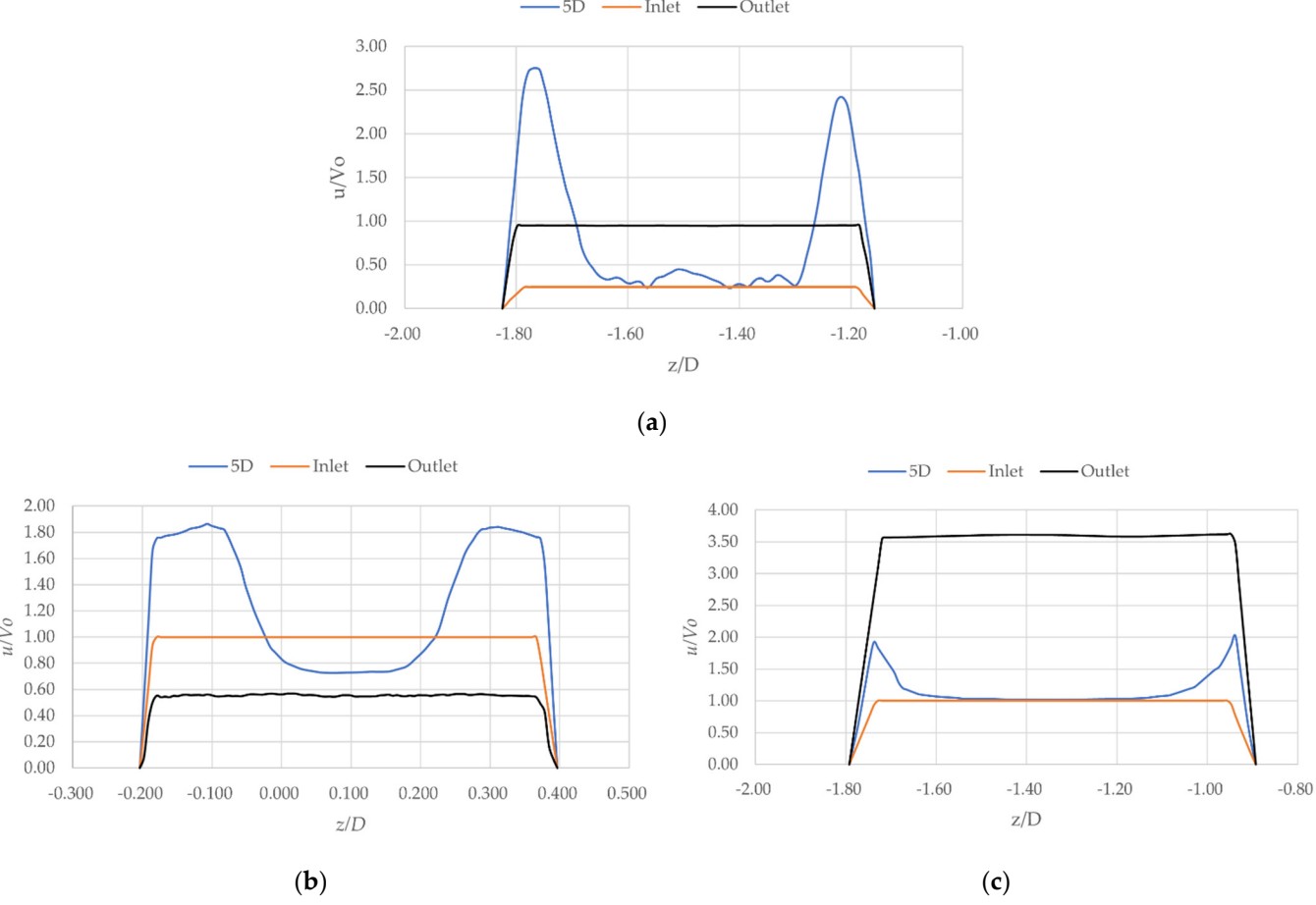

**Figure 7.** Streamwise velocity profiles of a multiple frame reference (5th frame) at time 0.112 s: (**a**) three blades, (**b**) four blades, and (**c**) five blades.

Figure 8 shows a numerical simulation, particularly in the near wake of the propellers. In the streamwise *z*-direction, where $\omega'$, is the inlet vorticity and $\omega$ is the vorticity in the *x*-direction. All propeller cases' wake maxima at the outer radii associated with tip vortices, as well as with their high and low vorticity peaks and where there should be relatively flat regions, are captured. It is very apparent that the four-blade propeller does not have a plateau region but shows two spikes. It can be attributed to hub vorticities spiking at the low rotational speeds as they appear in the hub region exhibiting stronger hub-vortex cavitations [4,35,36] compared to three- and five-blade propellers, noting that vorticity profiles strongly depend on the precise location of the coherent structures within the wake.

The vorticity fields feature much larger gradients than the streamwise velocity fields and significantly contribute to radiated noise levels.

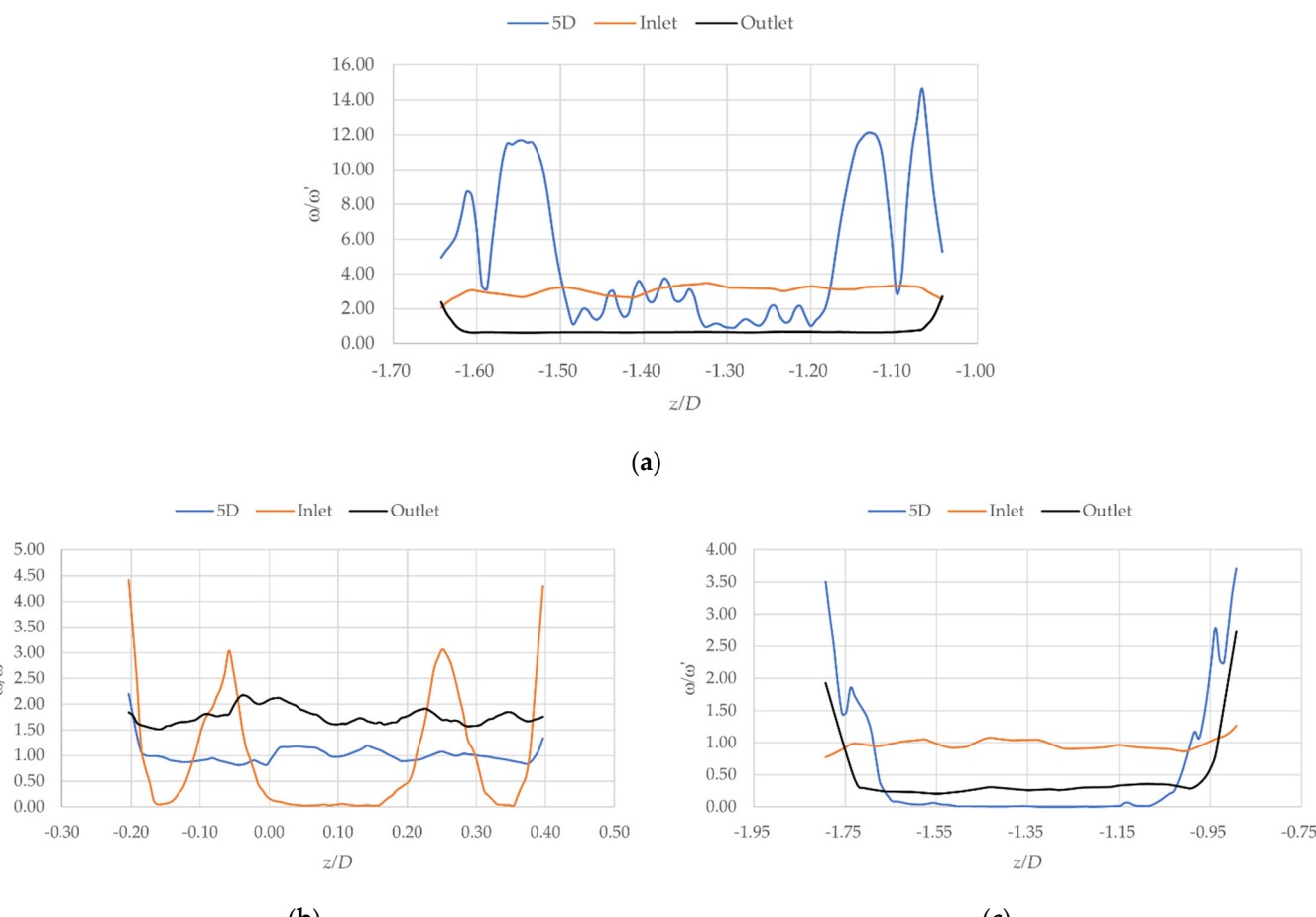

**(a)**

**(b)**

**(c)**

**Figure 8.** Relative vorticity profiles at time 0.112 s: (**a**) three blades, (**b**) four blades, and (**c**) five blades.

Two hydrophones (receivers R1 and R2) are used in the numerical study, and their positions are shown in Figure 2, with their results for the three propellers presented in Figure 9. The reference level in the solid body is considered a source of the sound where the blade surfaces.

The FLUENT's fast Fourier transform (FFT) function is used to convert time history signals to the frequency domain. The numerical approach simulates the occurrence of sheet cavitation on blade surfaces. There is sheet cavitation on blade surfaces, while cavitation and the edge vortex exist downstream and close to Hydrophone 2 (R2). According to the inverse-square-of-distance law, the overall SPL decreases with the increasing distance from the sound source for all blade types, as shown in Figure 9.

In Figure 10, the variations of sound pressure levels of the three propellers are combined and compared. It is seen that the noise difference between the three and four blades is no more than 10 dB. However, they are greater than 60 dB for the frequencies where the propellers produce the most noise. Therefore, blade propellers with an even number of blades significantly reduce noise. Furthermore, blade propellers with many even-number blades for the same dynamic conditions as the advancing ratio are quieter.

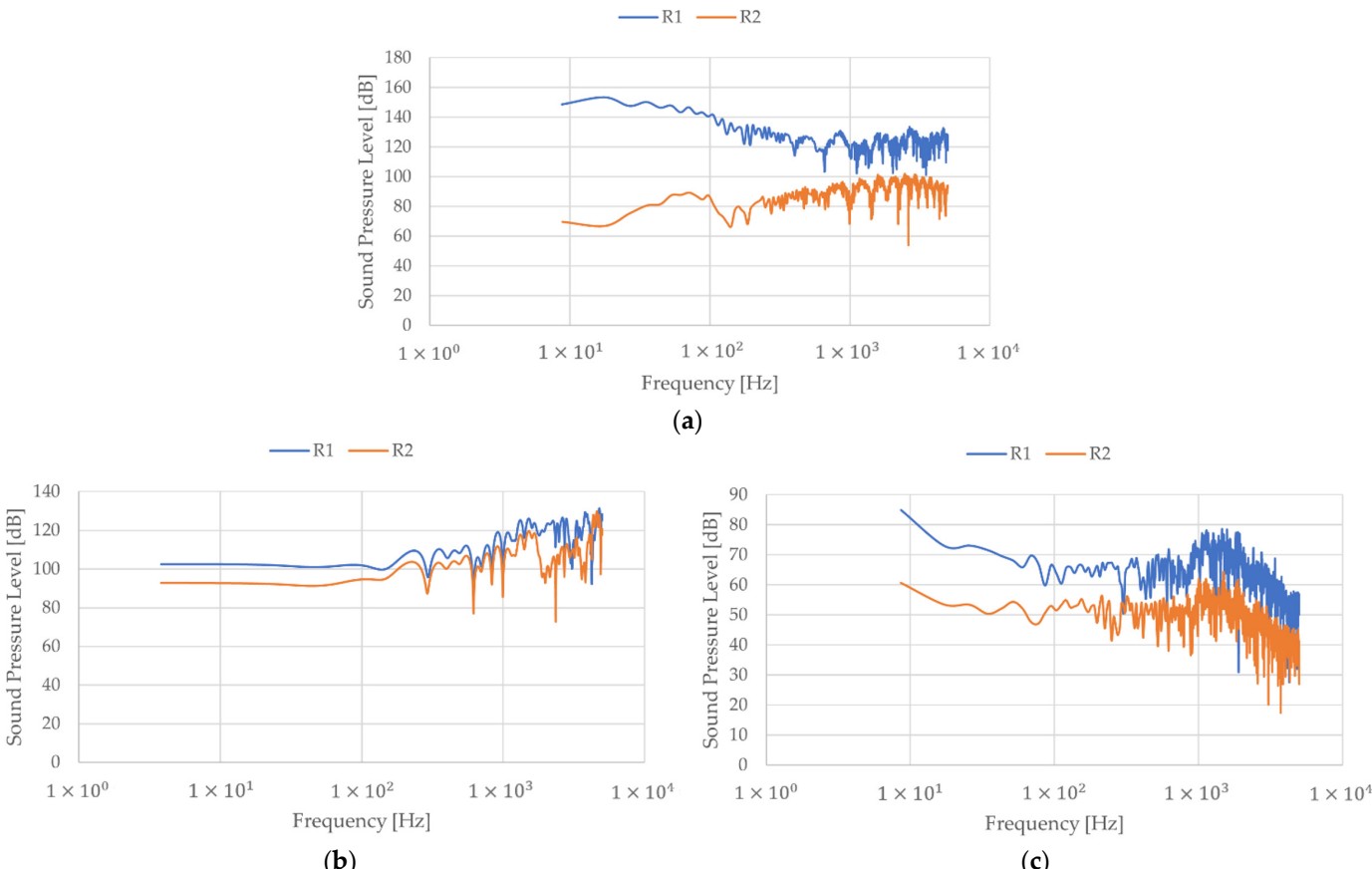

**Figure 9.** Sound pressure levels of (**a**) three blades, (**b**) four blades, and (**c**) five blades.

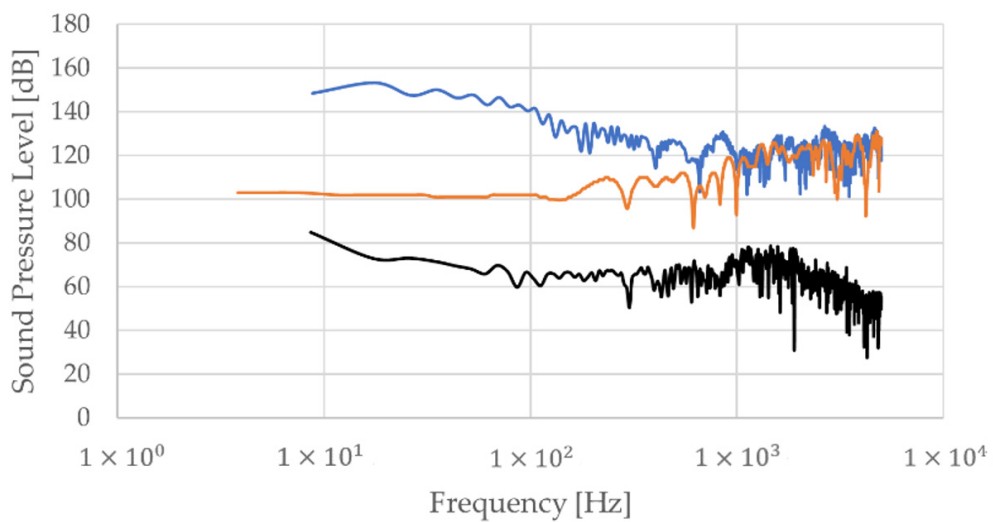

**Figure 10.** The contrast of sound pressure levels of all blades (3b, 4b & 5b) at hydrophone (receiver, R1 at 5d distance).

## 5. Conclusions

The Ffowcs Williams and Hawking's acoustic model, coupled with the Zwart et al. cavitation model, is employed to investigate the cavitating flow and noise acoustics of three types of marine propellers operating at a low rational speed. Some main conclusions are as follows:

- Tip-vortex cavitation:

It is possible to forecast the amount of radiated noise when both tip-vortex cavitation and leading-edge suction-side sheet cavitation are present. The inception and propagation of cavitation noise on a fixed advancing coefficient and the same skew angle are evident on the three-blade propeller at high frequencies. A high angle of incidence between the blade's leading edge and the water flow direction typically causes the hub vortex to cavitate.

- Vorticity:

The vortex sheets coming from the blades are defined by the distribution of sources and free vorticity over the blades and in the propeller's wake. Therefore, the performance could be enhanced by reducing the amount of vorticity shedding at least on the blade loading and geometry designs. Additionally, the roll-up of the cavitating tip vortex was closely related to propeller noise.

- Sound pressure level:

The sound pressure levels of the three propellers were compared. It follows that blade propellers with even fewer blades significantly increase noise. Additionally, for the same dynamic conditions as the advancing ratio, the blade propellers with even more blades are quieter.

- Geometry:

The number of blades significantly reduced the sound-pressure levels for equal power consumption at the same blade angle. The fluid follows the blades more closely, and the blade loading is reduced as the number of blades increases. Higher blade loadings cause the flow to concentrate on the pressure surfaces and create an exit velocity gradient. The blade's geometry determines the overall form of the pressure distribution around its surface for a given flow incidence.

One drawback of this study was the limitation on the number of mesh elements, such as the adaptive mesh refinement methods (applied near the propeller) that could be used. It is understood that the transient flow fluctuations and separation would have been better modeled and presented. As a result, our study used ILES but was constrained by the strict mesh and calculational requirements. The sheet and cloud cavitation's temporary structures will be captured, and the evolution characteristics of the cavities and vortex structures will be thoroughly examined.

**Author Contributions:** Conceptualization, K.M.D. and T.J.K.; methodology, K.M.D.; software, validation, V.T.H.; formal analysis, T.J.K.; investigation K.M.D.; resources, V.T.H.; data curation, K.M.D.; writing—original draft preparation, K.M.D.; writing—review and editing, V.T.H.; visualization, supervision, T.J.K.; project administration, T.J.K. All authors have read and agreed to the published version of the manuscript.

**Funding:** This research received no external funding.

**Data Availability Statement:** Not applicable.

**Acknowledgments:** The authors acknowledge the help from the Mechanical and Industrial Engineering Technology department of the University of Johannesburg.

**Conflicts of Interest:** The authors declare no conflict of interest.

## Appendix A. ANSYS Workbench Parameterization

**Figure A1.** ANSYS workbench parameterization.

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
