# Peer review of "Numerical Modeling of Cavitation Rates and Noise Acoustics of Marine Propellers"

_mca, doi:10.3390/mca28020042_

Round 1

Reviewer 1 Report

The paper investigates cavitation phenomena using CFD modelling. The main problem is that such complex phenomena cannot be described in 3D with less than 500 000 elements, which is the limit of student version authors use. If the authors do not have appropriate tool for investigating problem of interest, they need to consider problems which can be described in 2D, so maybe 500 000 elements could be sufficient or consider using OpenFOAM, or some other software which is free.

Limit of number of elements can be appropriate justification for student papers/thesis, however this is not appropriate  justification for research article.  Appropriate validation of results,  grid independence study needs to be conducted so findings from the paper can have any credibility. Therefore the recommendation is reject.

Reviewer 2 Report

On the whole, it is a scientific, complete, research paper. The article meets the journal scope. The reviewer has some questions for the authors and some suggestions for revision.

1) How to understand the "rates" in title?

2) Figure 6: it is not clear for displaying, why not use iso-surface of vapor phase?

3) The authors are required to provide mesh independence test.

4) Is the sound source surface you specify the rotation domain or the blade surface?

5) What is the basis for the designation of time-step ?

Reviewer 3 Report

Brief summary:

The study numerically investigated noise dissipation, and cavitation produced by marine propellers. The findings might be interesting, but they should be more convincing, the structure of the manuscript must be improved and the authors should choose better the data for validation and verification of their model. English is quite good, but typos have occurred too much.

Broad comments:

There are some problems with references:

Reference 1 is wrong, there is no title, right brackets are missing, it should probably be: Bertschneider, H., et al. "Specialist committee on hydrodynamic noise." Final report and recommendations to the 27th ITTC. Copenhagen, Sweden 45 (2014)

Reference 7 is nonsense, it should be cited as an on-line source.
IMO Guidelines for the Reduction of Underwater Noise
https://www.nrdc.org/resources/imo-guidelines-reduction-underwater-noise

References 16, and 23 were not cited at all.

Reference 22 was cited as late as on line 199, i.e. after references 24-26, numbering of references should be corrected.

Abstract: The abbreviation FHWH must be changed to FW-H. The abstract as a whole is misleading, it promises information that is not discussed in the paper, it should be revised completely.

The keyword “Ffowcs Williams-Hawkings” cannot be used, it must be completed by “model” , “equation” or “analogy”.

The introduction seems to be comprehensive, and it acquaints readers quite well with the current state of knowledge in the field. There are some formal details that should be improved (see the Specific comments).

l.33-37 There is no need to explain what “cavitation” is, the sentences should be moved to the theoretical part and here it should be replaced by “One of the most challenging problems is cavitation. When vapor cavities start to form, ….”

The theoretical part is not sufficient and there are some serious errors that must be fixed. First of all, it is not wise to mix governing equations and LES simulation into one subchapter. Some theoretical introduction describing the problem as a whole from the physical point of view should be beneficial.

The chapter Numerical methods might follow, but a brief explanation of the principles of LES simulation is necessary, including introducing explicit and implicit filtering.

There is one more essential problem – there are two fluids that are studied in the presented research and the authors should specify exactly which of them they are just dealing with. The first one is water – it certainly could be counted as incompressible, therefore, eq. 1 and 2 should be valid (but they are mentioned neither in reference [1] nor in [2], some other reference should be cited). The density r, however, certainly cannot be the air density; the letter p usually denotes pressure, not pressure fluctuation; concerning eq.(2) it is more often to write the first term of the right-hand side of the equation on the left-hand side.

The authors should explain the reason, why they use different equation notations in the eqs. (1)-(2) and in the eqs. (3)-(8).

The subchapter describing cavitation is insufficient, it is necessary to start with information that the governing equations are valid for the mixture fluid density and introduce al and av.

CFD methodology provides the most important data about the modeling. The authors should explain the reasons which led to their choice of parameters (geometric parameters of the propeller and the working conditions)

Model validation is not convincing much, the authors compare their results with experimental findings that have not much in common with their conditions. The figure 4 is unacceptable, there are too many points from the current study and nothing usable can be seen.

The conclusion part should be reduced and just the most important findings should be presented, highlighting what is new and beneficial is necessary

Specific comments:

60 the full meaning of CFD should be moved to line 57, where it is used for the first time; there is no need to use it on line 85, the abbreviation should be enough

70, 94, 382 abbreviation LES is sufficient, the full meaning was introduced above

66, 223 abbreviation RANS is sufficient, the full meaning was introduced above

79         “When imitating cavitating, unsteady flows.” should be joined with the previous sentence

97         continuity equation (singular is necessary)

107       eq. (4) The first term denotes partial time derivative, the subscript must be “t” not “s”

112       excessive occurrences of  “SGS”

155       brackets are missing, Dirac function should be written as δ(f)

167-9    there are several formal errors (typos) in eqs. (17) and (18):

c0 in the first exponential function in both equations must be corrected (zero is subscript)

Ф in equation (17) should be corrected to j, “mBMa” should be written in italics in eq. (17)

s in summation in eq. 18 ranges from -a not minus infinity

193       there are just three cited references [24-26], this cannot be called “many”, “several” is more appropriate

229       LES and acoustic pressures were not measured, they were evaluated

292       “cavity” correct to “cavitation”

300-2    the last sentence is quite incomprehensible

310       voracity” correct to “vorticity”

355       “proper” correct to “propeller”

363       the sentence sounds strange

Reviewer 4 Report

l. 79: "When imitating cavitating, unsteady flows [13]." - unfinished sentence

l. 127: What is C_S in the SGS eddy viscosity equation (9)?

l. 134: bubble's expansion and deflation - eq. (11) - deflation is in fact bubble collapse and is not governed by re-condensation rate 

l. 205: What are the boundary conditions for the acoustic waves leaving the cylindrical CFD domain?

l. 206, Table 1: What are "Skew angle" and "Cavitation coefficient"?  How does this coefficient fit into eq. (11) - (18)?  "Blade Pass Frequency (BPF) noise level intensity [Hz]" does nor make sense.

l. 209: -->quieter, not "quitter"

l. 223: "using a RANS ... solver" contradicts l. 87 "made use of the LES method".

l. 224: "Schnerr-Sauer cavitation model" or "cavitation model by Kubota et al." as in l. 133?

l. 229: What is this "flow and acoustics" solver? How are the "LES and acoustic pressures" obtained?

l. 308: "an excellent numerical simulation" - What is the evidence for this?

Round 2

Reviewer 1 Report

The authors did not provide satisfactory mesh sensitivity study. The authors state that they use parametrization module, but the authors only change element size in domain without entering workbench (essentially it is only way of obtaining data/manipulation with simulations). The authors only change element size in domain. Data provided in Appendix A is not visible.

The figure 3 which is provided only shows that mesh independence is not obtained, indicating that the results are not valid. The authors investigate complex cavitation phenomena, which cannot be appropriately described with student version. For example, paper published in 2012 used 1.3 milion of elements for roughly 2 m length and 1 m diameter. The authors of this paper used 500 000 elements for domain longer than 8 m and more than 2 m diameter. Therefore, the recommendation for this publication is unfortunately reject.

Subhas, S., Saji, V. F., Ramakrishna, S., & Das, H. N. (2012). CFD analysis of a propeller flow and cavitation. International Journal of Computer Applications, 55(16).

Reviewer 3 Report

Brief summary:

The study numerically investigated noise dissipation, and cavitation produced by marine propellers. The findings are interesting, the authors honestly explained the research weaknesses, and the manuscript is certainly worth publishing, supposing that the remaining errors are fixed. English is fine.

Broad comments:

There are some problems with references:

Reference 1 is wrong, there is no title, right brackets are missing, it should probably be: Bertschneider, H., et al. "Specialist committee on hydrodynamic noise." Final report and recommendations to the 27th ITTC. Copenhagen, Sweden 45 (2014)

Reference 7 is nonsense, it should be cited as an on-line source.
IMO Guidelines for the Reduction of Underwater Noise
https://www.nrdc.org/resources/imo-guidelines-reduction-underwater-noise

Reference 24 (Lighthill) was not cited at all, although the author is mentioned twice in the text.

Reference 23(22) was cited as late as on line 182(199), i.e. after references 25-27 (24-26), the numbering of references should be corrected.

Abstract: is OK

The keyword OK

The introduction was improved very well, therefore no other corrections are necessary because the new version sufficiently explains all weak points.

The conclusion part is long and seems to be rather featureless, maybe some bulleting should help to make it more attractive.

Specific comments:

68(79)   “When imitating cavitating, unsteady flows.” should be joined with the previous sentence NOT CORRECTED

81-82    In the same cavitating conditions, were used to perform grid sensitivity tests to bring our numerical results to a converged state.  Grammar fault - the subject is missing in the sentence.

150       check the eq.(15), “ϕ” is necessary, not “j”

163       ?ref is the reference acoustic pressure and 10−6 Pa for water.

185,290              excessive dot at the end of the sentence

218       the redundant line must be deleted

300-2    the last sentence is quite incomprehensible NOT CORRECTED

308(310)            voracity” correct to “vorticity” NOT CORRECTED

353(355)            “proper” correct to “propeller” NOT CORRECTED

Round 3

Reviewer 3 Report

The manuscript seems to be corrected sufficiently.

My best regards